# Evaluation of the Color Stability, Water Sorption, and Solubility of Current Resin Composites

**DOI:** 10.3390/ma15196710

**Published:** 2022-09-27

**Authors:** Wenkai Huang, Ling Ren, Yuyao Cheng, Minghua Xu, Wenji Luo, Desong Zhan, Hidehiko Sano, Jiale Fu

**Affiliations:** 1School and Hospital of Stomatology, China Medical University, Liaoning Provincial Key Laboratory of Oral Disease, Shenyang 110002, China; 2Department of Dental Materials Science, School and Hospital of Stomatology, China Medical University, Shenyang 110002, China; 3Department of Restorative Dentistry, Division of Oral Health Science, Faculty of Dental Medicine, Hokkaido University, Kita 13, Nishi 7, Kita-ku, Sapporo 060-8586, Japan

**Keywords:** resin composite, color stability, water sorption, solubility

## Abstract

This study aims to assess the color stability, water sorption, and solubility of 11 resin composites as commercially available dental products. Twenty samples (10 mm in diameter and 2 mm in thickness) of each material were fabricated using a customized silicone mold, followed by immersion in each of curry, coffee, wine, and distilled water for 28 days (*n* = 5). Baseline shade and color changes (ΔE) were measured using a reflection spectrophotometer. The CIE L*, a*, b* system was used to evaluate the color changes. Five samples of each resin composite were applied to test water sorption and solubility according to ISO 4049:2009. As a result, the ∆E values were significantly influenced by each of the three factors (composition of material, solution, time) and the interactions between them (*p* < 0.001). Highest resistance to discoloration was achieved by Ceram.X One Universal (CXU), followed by Magnafill Putty (MP). Generally, microhybrid composites showed fewer color changes than nanohybrid composites and giomers. DX. Universal and Filtek Z350 XT showed the highest ΔE values in all colorants. All materials tested in this study fulfilled the criteria of ISO 4049:2009; CXU and MP had the lowest water sorption and solubility. The Pearson test showed statistically significant positive correlations between water sorption and ΔE and between solubility and ΔE.

## 1. Introduction

Resin-based composites have become the primary choices for esthetic restoration, due to their advantages of operational convenience, favorable physical and chemical properties, an extensive range of available colors, and excellent biological compatibility, as well as the capability of recovering the natural appearance of teeth [1,2,3,4]. However, these resin-based composites are susceptible to discoloration as time passes, owing to intrinsic and extrinsic factors. The extrinsic factors include the intensity and duration of polymerization, and exposure to environmental factors such as ultraviolet radiation, heat, moisture, and food-derived colorants. With regard to the intrinsic factors, the characteristics of the filler particles and the compositions of the resin matrices influence the smoothness of the cured surfaces of composites, which in turn affects the susceptibility to external colorants [5].

Humans consume many foods and drinks containing colorants in their daily lives. For example, Indian food contains a large number of different ingredients, so it has a high staining capacity [6]. Many studies have also demonstrated that resin composites can readily undergo discoloration when in contact with solutions such as coffee, tea, and wine [7]. Moreover, beverages may lead to deterioration of the physical properties of resin composites, thereby decreasing their restorative qualities [7].

Resistance to discoloration is an important standard reflecting the success of resin restoration [8]; esthetics has been cited as one of the most relevant reasons for replacement of anterior resin composite restorations [9]. Among people with high esthetic expectations, neither a color change of restorations nor frequent surface polishing or renewing surfaces is acceptable [3]. However, limited information is available from manufacturers about the susceptibility of resin composites to staining; there is also a lack of scientific data to illustrate their clinical color stability [3,10].

Water sorption also plays a crucial role in the clinical success of dental materials. Ideally, resin composites should be highly stable and impermeable to water, but the polymer network can absorb moisture, which can add several percentages to the material’s total weight [8]. The absorption of water has many negative effects on restorative materials, such as color change, impairing of the mechanical properties, decreasing of the wear-resistance, hydrolyzing and degrading of chemical bonds at the resin–filler interface [8,11]. In addition, water absorption produces the external movement of residual monomers and ions, consequently increasing the solubility of resin composites, which can degrade the biocompatibility of materials, reduce their volume, weaken their mechanical properties, and lead to discoloration by chemically degrading the filler–resin bond of the resin matrix [8,11].

This in-vitro study aims to compare the color stabilities of 11 current composites commercially available and measure their water sorptions and solubilities. The null hypotheses in this study were as follows: (1) Resin material, immersion solution, duration of immersion and the interactions among them have no influence on staining intensity. (2) There are no significant differences in water sorption and water solubility among the 11 types of resin composites.

## 2. Materials and Methods

### 2.1. Color Stability

#### 2.1.1. Sample Preparation

The 11 resin composites (A1 shade) used in this study include nanofilled composites, microhybrid composites, nanohybrid composites, nanoceramic composites, and giomers; the specific information for each resin composite is shown in Table 1. Twenty samples with 2 mm in thickness and 10 mm in diameter for each material were made using a customized silicone mold [12,13]; the sample size was determined by previous studies [14,15]. Samples were fabricated by filling materials into the silicone mold. To exclude extra material and obtain a smooth surface, a polyester strip and a 1-mm-thick glass slide were compressed onto the top of the material [12,13]. Then, the top surfaces of each sample were light-cured by a light-curing unit (Kerr Demi Plus; Kerr, CA, USA) for 40 s under the standard curing mode [2]. The output irradiance of the light-curing unit was 1500 mW/cm^2^. During polymerization, the unit was always kept vertical and in contact with the glass slide. After polymerization, samples were removed from the mold and marked on the rough surface to distinguish the two surfaces. Finally, all samples were rinsed with water and air-dried, after which they were kept in distilled water for 24 h at 37 °C to ensure complete polymerization [16]. Subsequently, the colors of each air-dried specimen were measured as the baselines.

#### 2.1.2. Discoloration in Different Solutions

To compare the ability of the different materials to avoid discoloration in the various solutions, 20 samples from each material were randomly divided into four subgroups (*n* = 5), which were immersed in curry (commercial curry sauce; Laide Miao, Guangzhou, China), coffee (bottled American coffee; Costa, London, UK), wine (Great Wall, Beijing, China), and distilled water. The distilled water was used for the negative control group. Samples in each subgroup (*n* = 5) were individually immersed in vials containing 30 mL of staining solution and stored in an incubator at 37 °C for 28 days; the solutions were renewed every seven days to prevent contamination with bacteria and yeast. After immersion, the vials were shaken to ensure that the samples were fully exposed to the staining solutions.

#### 2.1.3. Color Change Assessment

After 24 h of immersion in distilled water at 37 °C following sample production, the specimens were removed and dried at room temperature. To avoid the influence of light in the environment, each sample was put in a dark box covered with a white background. The baseline color was measured from the smooth surface of each sample using a reflection spectrophotometer (Crystaleye; Olympus, Tokyo, Japan). The color evaluation was based on the Commission Internationale de l’Éclairage L*, a*, b* (CIELAB) color space system, in which L represents brightness (0, black; 100, white), and a and b are green–red and blue–yellow axes, respectively. Measurements of each sample were repeated three times, from which the mean values of L*, a*, and b* were calculated and recorded as L_0_, a_0_, and b_0_, respectively. The spectrophotometer was calibrated before each measurement with reference to the manufacturer’s recommendations.

The color measurements were performed at 1, 7, 14, 21, and 28 days after immersion in the staining solutions. All procedures were consistent with the method used for measurement at baseline and were completed by a single operator. L*, a*, and b* values of each sample were measured three times, from which the mean L*, a*, and b* values were calculated as L_x,_ a_x,_ and b_x_, where x represents the duration of immersion. The color changes were calculated using Equation (1):(1)$$\Delta E_{x}=\sqrt{{\left(L_{x}-L_{0}\right)}^{2}+{\left(a_{x}-a_{0}\right)}^{2}+{\left(b_{x}-b_{0}\right)}^{2}}\;$$

The average ΔE was recorded as $\Delta E_{1}$, $\Delta E_{2}$, $\Delta E_{3}$, $\Delta E_{4}$, and $\Delta E_{5}$, representing the extent of discoloration for each group after 1, 7, 14, 21, and 28 days of staining, respectively.

### 2.2. Water Sorption and Solubility

A total of 55 samples (1 mm thickness × 15 mm diameter) were made (five samples of each material) using a Teflon ring mold, according to ISO 4049:2009 [17]. After fabrication, peripheral irregularities of samples were removed with 1000-grit silicon carbide paper. Subsequently, the samples of each group were transferred to vials and stored in a desiccator (DHG-9240A; Shanghai Yiheng Technical Co., Ltd., Shanghai, China) at 37 ± 1 °C. Anhydrous self-indicating silica gel was placed on the bottom of the desiccator. Twenty-two hours later, samples were placed into another desiccator at 23 ± 1 °C for 2 h. Then, the samples were weighed using a digital balance (ES1035A; Deante Sensing Technology Co., Ltd., Tianjin, China) with accuracy of 0.1 mg. This process was repeated until the decrease in mass between two consecutive measurements was less than 0.1 mg, at which point the final weight was recorded as m_1_. After drying each sample, two diameters perpendicular to each other were measured by an electronic digital caliper with 0.01 mm accuracy and their mean was calculated. The thickness of the sample was also measured at its center and at four points at equal intervals along its circumference. The average diameter and thickness were used to calculate the volume (V).

Each sample was separately placed into 10 mL of distilled water in a marked incubator at 37 ± 1 °C for seven days. Subsequently, the samples were removed, rinsed in water, and thoroughly dried with absorbent paper, and their weight was determined as m_2_. To calculate solubility values, the dehydration procedure was repeated until the samples acquired a stable weight, which was recorded as m_3_.

The water sorption and solubility (μg/mm^3^) of each material were calculated using Equations (2) and (3) as follows:(2)$$\mathrm{Water}\;\mathrm{sorption}=\frac{m2-m3}{v}$$
(3)$$\mathrm{Water}\;\mathrm{solubility}=\frac{m1-m3}{v}$$

### 2.3. Statistical Analysis

Three-way analysis of variance (ANOVA) was used to evaluate the factors (resin materials, staining solutions, and storage time) and their correlative effects on color change (ΔE) using the statistical software SPSS (version 26; Statistical Package for the Social Sciences, Chicago, IL, USA). The Kolmogorov-Smirnov test and the Levene test were used for checking normality and homogeneity. For between-groups comparison of color stability, one-way ANOVA was used to evaluate the effects of materials and solutions, and the effects of immersion time were evaluated by repeated measures of ANOVA. For water sorption and solubility, one-way ANOVA was chosen. Tukey’s HSD test was performed for comparisons of mean values when equality of variances between groups was verified; otherwise, it was replaced by the Games-Howell test. Possible correlations between water sorption and color change, solubility and color change, and water sorption and solubility were determined by Pearson’s correlation test. The value of α = 0.05 was recognized as indicating a significant difference in the above statistical tests.

## 3. Results

Three-way ANOVA revealed that the three factors (type of material, type of solution, and immersion time) and the interactions between them all significantly influenced the ΔE values (*p* < 0.001). According to the Kolmogorov-Smirnov test, all of the data were normally distributed, and therefore parametric tests were performed.

### 3.1. Color Stability of Resin Composites

The changes in color over time of the 11 resin composites immersed in the different colorants are shown in Figure 1. Overall, curry demonstrated the strongest staining capacity for all materials, followed by wine, coffee, and then distilled water. In the curry, coffee, and wine groups, the ∆E values exhibited gradually increasing trends over time for all materials. The ∆E_2_ values of each sample were significantly higher (*p* < 0.05) than the ∆E_1_ values, with the exception of F03 in curry. After staining for 21 days, about half of the samples did not develop any obvious color changes at the subsequent timepoint (*p* > 0.05). In the water group, the color change was irregular among the time points, but the mean ∆E values for all materials did not exceed 3.3. The mean values and standard deviations of ΔE at each timepoint after immersion of the resins in the four different staining solutions are presented in Table 2.

As for the resin samples, Z350 and DXU performed intensive susceptibility to staining from seven days of immersion. After 28 days, DXU underwent the strongest discoloration in curry (∆E_5_ = 65.05) and coffee (∆E_5_ = 24.97), followed by Z350 (∆E_5_ = 61.22 in curry, ∆E_5_ = 21.19 in coffee). Z350 was the most susceptible to wine (∆E_5_ = 37.76), followed by DXU (∆E_5_ = 36.32). B2 (∆E_5_ = 20.32 in coffee, ∆E_5_ = 26.63 in wine) and F03 (∆E_5_ = 19.55 in coffee, ∆E_5_ = 26.41 in wine) were also intensively stained by coffee and wine, but were rather resistant to discoloration in curry. F03 showed significant resistance to discoloration in curry (∆E_5_ = 48.22) compared with the other materials (*p* < 0.05) and presented the lowest ∆E values at all immersion times. CXU underwent the third lowest color change in curry (∆E_5_ = 55.76), and the lowest in coffee (∆E_5_ = 11.91) and wine (∆E_5_ = 13.52). MP performed similar staining resistance to CXU in these solutions, although it differed significantly (*p* < 0.05) from CXU when stained by wine. CD, CS, TNC, DF, and Z250 exhibited intermediate capacities to avoid discoloration. In particular, CS was associated with greater staining resistance (∆E_5_ = 54.12) in curry than the other materials, with the exception of F03. DF remained rather stable in coffee (∆E_5_ = 12.85) and wine (∆E_5_ = 19.06). Z250 had lower ∆E_5_ values than CD and TNC in curry and coffee, and higher ∆E_5_ values in wine, but the differences between them were not statistically significant (*p* > 0.05). Figure 2 illustrates the staining situations of four representative material samples in each staining solution.

### 3.2. Chromatic Parameter Changes

Chromatic parameters contributing to ∆E_5_ are shown in Table 3. There was a significant positive change in ∆b for all materials immersed in curry, indicating a dramatic shift to a yellow color. The dominant change in color of the composites in coffee and wine can be ascribed as a shift to negative ∆L values, meaning that the specimens became darker. In the control group, the changes of ∆L, ∆a, and ∆b values were not obvious and were irregular. The results are further illustrated in Figure 3.

### 3.3. Water Sorption and Solubility

Table 4 and Figure 4 present the mean values and standard deviations of water sorptions and water solubilities of the tested materials. Z350 had higher water sorption values than the other materials (*p* < 0.05), followed by F03 and BT. MP showed the weakest water sorption (*p* < 0.05), followed by CXU. Z250, DXU, and TNC had similar water sorption values (*p* > 0.05). The water solubility values of CD and CS were significantly higher than those of the other materials (*p* < 0.05). There were no significant differences (*p* > 0.05) in water solubility values of materials, with the exception of Z350, CD, and CS. The Pearson analyses showed a positive statistically significant correlation between color changes and water sorptions (r = 0.313, *p* < 0.05), as well as between color changes and solubilities (r = 0.271, *p* < 0.05). A positive correlation between water sorption and solubility was observed, but there was no statistical significance (r = 0.203, *p* = 0.136).

## 4. Discussion

Resin composites have become indispensable parts of contemporary esthetic restorative treatment. However, resin restorations are at risk of discoloration as they are regularly exposed to a variety of food- and drink-based colorants. The intrinsic factors affecting the degree of discoloration include the type of resin matrix, as well as the type, size, and amount of inorganic filler [18]. Meanwhile, water sorption can weaken the bonds between the resin matrix and the filler particles, subsequently inducing microcracks or interfacial gaps at the resin–filler interface, through which the staining solutions can pass and contribute to the color change of resin composites [19]. Therefore, the aim of this study is to compare 11 different resin-based composites commercially available for use in dentistry with regard to their color stability, water sorption, and solubility.

The staining intensity was also reported to be affected by the surface roughness of resin composites [20]. Therefore, in this study, polyester strips were used to standardize the roughness and create a smooth surface when pressing the glass plates against material surfaces, resulting in a surface rich in the resin matrix and without an oxygen inhibition layer [21]. According to previous studies, this method without polish can simulate a severe clinical situation [12,22,23].

In the present investigation, all solutions were selected from commercial products available on the market. Curry, coffee, and wine were used as staining solutions because they are common food-based colorants in everyday life. Previous studies demonstrated the staining potential of these solutions [13,24,25]. Distilled water was used as a control, as it caused little color change in previous studies [26].

All tested samples were fabricated from the A1 shade, which is commonly used for young patients with higher expectations or esthetic requirements [5]. The period of staining in solutions lasted for 28 days, which is equivalent to around two years in the oral environment, according to the finding of Ertas et al. that one day of staining in vitro is equivalent to one month in vivo [27]. The baseline colors and color changes after staining were measured using a spectrophotometer and evaluated using the CIE L*, a*, b* system. Color changes (∆E) were calculated based on L, a, and b values. Studies have reported that ∆E values less than 1.5 are unperceivable by humans, while ∆E values of >3.3 are unacceptable in a clinical context [16,28,29].

Curry exerted the strongest staining among all tested colorants [25], which was consistent with the fact that it caused the highest ∆E_5_ values for all tested materials (Table 2). Moreover, after one day of immersion in curry, all samples showed ∆E values exceeding 3.3. In the coffee and wine groups, all samples showed clinically unacceptable color changes (∆E_2_ > 3.3) after seven days of immersion. In the literature, there are inconsistent findings about the intensity with which coffee and wine stain resin composites. In the current study, wine caused a more severe color change than coffee, which is consistent with one previous study [10]. In the control group, each material displayed an acceptable color change (∆E < 3.3) consistently. To sum up, the ranking of the staining intensity of the colorants, from most to least, was: curry > wine > coffee > distilled water.

The staining process lasted for 28 days, as shown in Figure 1, during which a gradually increasing trend of ∆E values over time was observed. As for chromatic parameters (Table 3 and Figure 3), positive ∆b values were dominant in the curry group for all materials, since curry contains a large amount of curcumin [30]. Chromaticity testing showed negative ∆L values for all tested materials and higher b values in most instances in the wine group. This trend was in agreement with a study by Falkensammer et al. [31]. The majority of color changes in the coffee group involved increasingly negative ∆L rather than positive ∆b, which contradicts previous studies [14]. One possible reason for this discrepancy is that the yellow colorants in coffee can be eluted by rinsing in water [32]. Besides, the bottled coffee utilized in this study differed from the instant coffee used in other studies. Taking the findings together, the color of different resin composites changed significantly when the immersion time was prolonged to 28 days, compared with the color change upon immersion in distilled water (Figure 1).

As for the effect of the particular resin material, it has been reported that lower ratios of weight and volume between inorganic fillers and resin matrix, larger filler particles, and hydrophilic resin matrix are risk factors for the discoloration of resin composites [33]. Based on the category of filler size, the 11 types of resin composites tested here were classified as follows: nanofilled (MP, Z350), microhybrid (Z250, CS, DF), nanohybrid (CD, TNC), nanoceramic (CXU), and giomers (B2, F03), as well as one brand of resin with no information on the filler type disclosed by the manufacturer (DXU).

Z350 is a nanofilled resin composite with small filler particles, intended to lead to a smoother surface and consequently less surface staining [34]. However, in the present study, Z350 showed high susceptibility to staining, which is similar to the findings of other studies [5,35]. One possible reason for this is that Z350 has a higher level of the hydrophilic monomers Bis-GMA and TEGDMA, resulting in more water sorption [35]. Moreover, Rodrigues et al. indicated that the presence of microporosities and the quantum effect can also promote water sorption and pigment retention [36], which may be the factors resulting in the marked changes in color of Z350 in the staining solutions.

As another nanofilled composite, MP was selected in the present study because it is a novel resin composite that has not been marketed globally, so its properties and color stability have rarely been reported. As shown in Table 2, MP had rather stable anti-staining performance in all staining solutions, although it showed a larger color change than CXU. Both Bis-MEPP and UDMA employed in the resin matrix of this material are hydrophobic monomers; Bis-MEPP is considered a rigid monomer which is more hydrophobic compared with UDMA [37,38]. MP was the only material containing Bis-MEPP in this study, so the high resistance to discoloration and low water sorption of MP may be related to this component.

In this study, the nanohybrid composites (CD, TNC) underwent greater discoloration than the microhybrid composites (Z250, CS, DF), which is in agreement with previous studies that demonstrated that composites with a smaller filler size do not necessarily exhibit better stain resistance [39,40]. As nanohybrid resin composites, both CD and TNC contain Bis-GMA, UDMA, and TEGDMA, while CD contains TCD-DI-HEA monomers, which can enhance the hydrophobic effect of UDMA [35]. However, in our study, CD performed similarly to TNC in all staining solutions, which might be attributable to these two materials having similar filler types and proportions. Z250, CS, and DF are microhybrid composites, of which Bis-GMA is a common component. Z250 is expected to be more stain-resistant than CS and DF due to the fact that it has an abundance of fillers (82% by weight) and hydrophobic components of its resin matrix, such as Bis-EMA and UDMA. However, Z250 did not exhibit better resistance to discoloration than DF and CS, except for showing lower ∆E values than DF in the curry group (*p* < 0.05). The fillers of Z250 were unsilanated, which may have led to a weak connection of filler and matrix, facilitating permeation of the colorants. Topcu et al. also found that Z250 is more susceptible to discoloration than CS [41].

The resin matrix of B2 (TEGDMA, Bis-GMA) was hydrophilic; the flowable composite F03 had the same resin matrix, filler type, and filler size as B2, while the filler content of F03 was even lower than that of B2, which can explain the susceptibility of B2 and F03 to staining in coffee and wine. However, F03 performed better than B2 in all staining media, which is in agreement with a previous study [26].

CXU is a nanoceramic resin composite with different resin matrices, including methacrylate-modified polysiloxane, polyurethane methacrylate, and Bis-EMA. Several factors may contribute to the color stability of CXU, such as the presence of the hydrophobic monomer Bis-EMA, a highly condensed hydrophobic polysiloxane backbone, and the smallest filler size (2.3 nm). Therefore, CXU exhibited strong resistance to discoloration in all tested solutions in the present study [42]. In summary, the first hypothesis that the three factors (resin material, immersion solution, duration of immersion) and the interactions among them have no influence on staining intensity was rejected.

Water sorption and solubility are variables influenced by multiple factors. According to ISO 4049, water sorption and water solubility values of resin composites should not exceed 40 μg/mm^3^ and 7.5 μg/mm^3^, respectively [17]; all materials tested in this study met these criteria. Z350 was associated with the highest water solubility, as it had rather large amounts of hydrophilic monomers. B2 and F03 are giomers with “pre-reacted glass ionomer” (PRG)-based fluoro-alumina-silicate glass, which can release and recharge fluoride by absorbing a certain amount of water. Besides, giomer composites might have more surface vacancies via the release of fluoride ions [43]. As a result, B2 and F03 exhibited rather higher water sorption values. In contrast, CXU and MP did not tend to absorb water because of their hydrophobic components and structures.

The absorption of water may expand the gap between polymer chains, allowing unreacted monomers to leach out. The water solubility of resin composites is determined by the structure of monomers, fillers, and the degree of conversion [44]. Both CD and CS contain barium aluminum fluoride glass; notably, CD and CS showed significantly higher solubility than the other materials. It has been reported that fluoride-releasing composites can be expected to have higher solubility, as they need water diffusion to occur to be effective [45]. However, materials with high water sorption do not necessarily demonstrate high solubility [44]. B2 and F03 could also release fluoride ions, but they did not demonstrate as high solubilities as those of CS and CD, which may be related to the particular S-PRG fillers. The acid-base reaction occurring in S-PRG fillers helped to establish a surface-modified layer that can protect the glass core when exposed to moisture [46]. As the above findings show, the second null hypothesis that the tested composites would exhibit no significant differences in water sorption and water solubility was also rejected. Particularly, Pearson analysis indicated that water sorption and solubility were closely related to the extent of discoloration, especially water sorption, which was consistent with the results of this study.

This work, involving an in-vitro experiment, had the following limitations: (1) different brands/types of curry, coffee, and wine may lead to different experimental results, but only one type of each was used in this study; (2) the actual conditions such as saliva and water washing, the real temperatures in the mouth of patients, and the effects of brushing were not simulated; (3) to imitate a more ideal clinical scenario, the effect of polish on staining intensity should be considered in further investigations; and (4) repolishing procedures after staining should be included to distinguish the intrinsic color changes of the specimens from the superficial surface staining [7]. Therefore, there is a need for further investigation of the actual performance of different resin composites in the oral cavity.

## 5. Conclusions

Discoloration is a multifactorial process, which is influenced by different colorant solutions and staining durations. The staining intensities of the solutions, from most to least, was as follows: curry > wine > coffee > distilled water.

The susceptibility to staining was also dependent on the structure and composition of the resin composites, with CXU and MP being most resistant to discoloration upon immersion in all tested colorants. Microhybrid resin composites presented less discoloration than nanohybrid resin composites and giomers in this study.

Water sorption and solubility were factors associated with discoloration process of resin composites. CXU and MP had lower water sorptions and solubilities than the other composites tested in this study.

## Figures and Tables

**Figure 1 materials-15-06710-f001:**
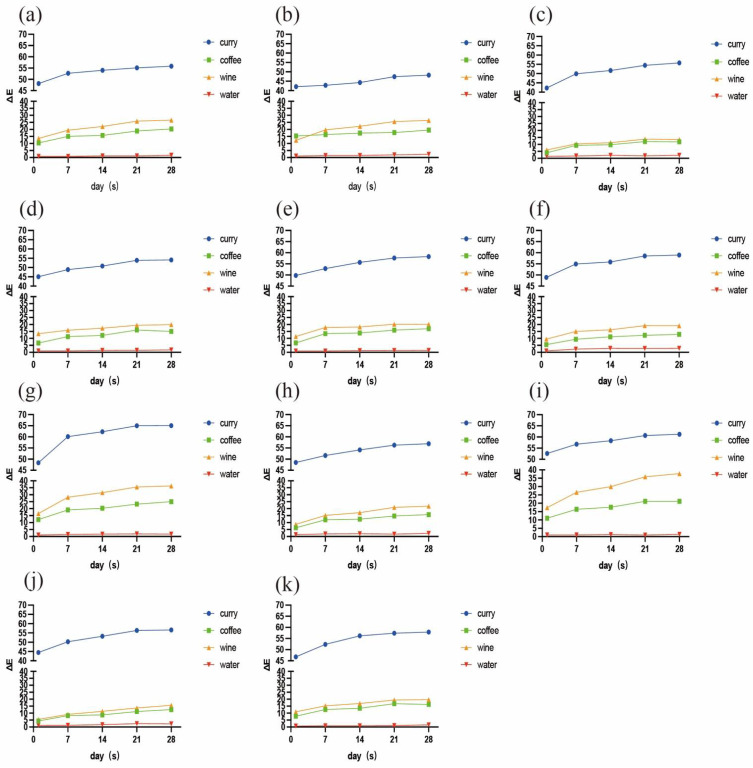
Color changes over time of 11 composite resins immersed in curry, coffee, wine, and distilled water for 1, 7, 14, 21, and 28 days: (**a**) B2; (**b**) F03; (**c**) CXU; (**d**) CS; (**e**) CD; (**f**) DF; (**g**) DXU; (**h**) Z250; (**i**) Z350; (**j**) MP; and (**k**) TNC.

**Figure 2 materials-15-06710-f002:**
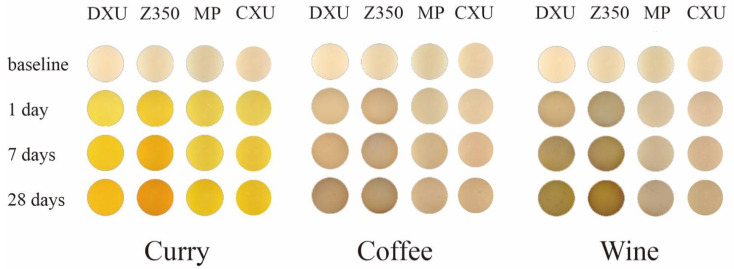
The staining situations of four representative material samples in each staining solution.

**Figure 3 materials-15-06710-f003:**

Radar graphs demonstrating CIE L*, a*, and b* value changes after immersion in colorants for 28 days: (**a**) curry (**b**) coffee (**c**) wine.

**Figure 4 materials-15-06710-f004:**
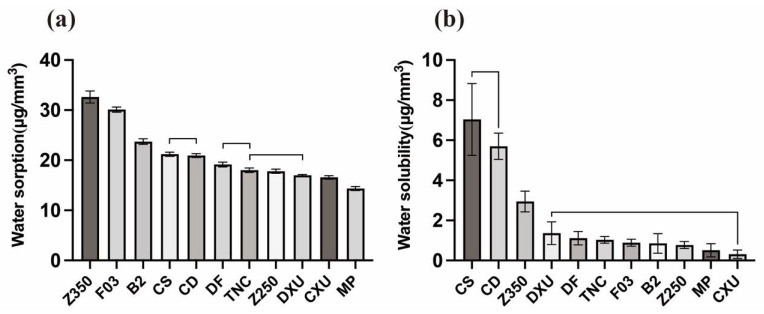
Bar graph illustrating mean water sorption and water solubility values of the composite resins. Groups underneath a line do not differ significantly: (**a**) water sorption (**b**) water solubility.

**Table 1 materials-15-06710-t001:** Summary of the products used in this study.

Material Abbreviation/Lot	Category	Resin Matrix	Main Fillers Type and Size	Filler Load (wt.%/vol.%)	Manufacturer
Beautifil II (B2/Lot: 072083)	Giomer	Bis-GMA, TEGDMA	S-PRG based on F-Br-Al-Si glass (0.01–4.0 µm)	83.3/68.6	Shofu Inc., Kyoto, Japan
Beautifil Flow Plus F03 (F03/Lot: 022147)	Flowable giomer	Bis-GMA, TEGDMA	S-PRG based on F-Br-Al-Si glass (0.01–4.0 µm)	66.8 by weight	Shofu Inc., Kyoto, Japan
Ceram.X One Universal (CXU/2103001285)	Nanoceramic	Bis-EMA, TEGDMA, methacrylate-modified polysiloxane, polyurethane methacrylate,	nanofillers (10 nm) and organically modified nanoceramic particles (2–3 nm), barium glass (1–1.5 μm) and ytterbium fluoride, camphorquinone, pigments	77–79/59–61	Dentsply, Konstanz, Germany
Charisma (CS/Lot: 71206)	Microhybrid	Bis-GMA, TEGDMA	Barium aluminum fluoride glass (0.02–2 µm) and colloidal silica (0.02–0.07 µm)	78/58	Heraeus Kulzer, GmbH, Hanau, Germany
Charisma Diamond (CD/Lot: K010054)	Nanohybrid	TCD-DI-HEA, UDMA, Bis-GMA, TEGDMA	Barium aluminum fluoride glass, and colloidal silica (5–20 µm)	81/64	Heraeus Kulzer, GmbH, Hanau, Germany
Denfil (DF/Lot: DF1376A1)	Microhybrid	Bis-GMA, TEGDMA	Barium aluminosilicate (≤1 µm), Fumed silica (0.04 µm)	80 by weight	Vericom, Anyang, Korea
DX.Universal (DXU/Lot: BJBJAE)	--	THFA, Bis-GMA, UDMA	Strontium glass, Silica (0.03–5 µm)	71 by volume	Dentex, Jilin, China
Filtek Z250 (Z250/Lot: NA53941)	Microhybrid	Bis-GMA, UDMA and Bis-EMA,	zirconia/silica without silane treatment (0.6 µm average)	82/60	3M ESPE, Saint Paul, MN, USA
Filtek Z350 XT (Z350/Lot: NA53296)	Nanofilled	Bis-GMA, UDMA, TEGDMA, Bis-EMA,	silica particles (20 nm), zirconia (4–11 nm), silica/zirconia clusters (0.6–10 µm)	78.5/63.3	3M ESPE, Saint Paul, MN, USA
Magnafill Putty (MP/Lot: 2108231)	Nanofilled	Bis-MEPP, UDMA	Barium glass	76–84/50–56	GC, Tokyo, Japan
Tetric N-Ceram (TNC/Lot: Z0VWJ)	Nanohybrid	Bis-GMA, UDMA, TEGDMA	Barium glass, ytterbium Trifluoride (40–3 µm)	80–81/55–57	Ivoclar Vivadent, Schaan, Liechtenstein

Bis-GMA: bisphenolglycidyl methacrylate; Bis-EMA: ethoxylated bisphenol-A dimethacrylate; TEGDMA: triethylene glycol dimethacrylate; UDMA: urethane dimethacrylate; Bis-MEPP: bisphenol-A ethoxylate dimethacrylate; THFA: Tetrahydrofurfuryl acrylate; TCD-DI-HEA: bis-(acryloyloxymethyl) tricyclo [5.2.1.02,6] decane; wt.%: weight percentage; vol.%: volume percentage.

**Table 2 materials-15-06710-t002:** Mean ∆E values and standard deviations of 11 composite resins after 28 days of staining.

Material	Solution	∆E_1_	∆E_2_	∆E_3_	∆E_4_	∆E_5_
B2	Curry	48.12 ± 1.82 ^A,a^	52.7 ± 1.38 ^A,b^	53.99 ± 1.55 ^A,c^	55.13 ± 1.05 ^A,d^	55.85 ± 1.35 ^A,e^
	Coffee	10.42 ± 1.29 ^B,a^	15.1 ± 1.63 ^B,b^	15.73 ± 1.25 ^B,b^	18.9 ± 1.15 ^B,c^	20.32 ± 1.22 ^B,d^
	Wine	13.65 ± 1.35 ^C,a^	19.45 ± 1.56 ^C,b^	22.06 ± 0.97 ^C,c^	25.88 ± 1.32 ^C,d^	26.63 ± 0.74 ^C,d^
	Distilled water	0.83 ± 0.15 ^D,a^	0.85 ± 0.2 ^D,a^	1.22 ± 0.28 ^D,b^	1.24 ± 0.33 ^D,^^b^	1.65 ± 0.66 ^D,^^c^
F03	Curry	42.13 ± 1.04 ^A,a^	42.80 ± 1.23 ^A,a^	44.28 ± 1.56 ^A,b^	47.40 ± 1.05 ^A,c^	48.22 ± 0.97 ^A,d^
	Coffee	15.38 ± 0.66 ^B,a^	16.35 ± 1.28 ^B,b^	17.38 ± 1.26 ^B,c^	17.75 ± 0.86 ^B,c^	19.55 ± 0.66 ^B,d^
	Wine	12.27 ± 0.55 ^B,a^	19.69 ± 0.99 ^C,b^	22.20 ± 0.12 ^C,c^	25.59 ± 1.49 ^C,d^	26.41 ± 1.52 ^C,d^
	Distilled water	1.04 ± 0.51 ^C,a^	1.53 ± 0.34 ^D,b^	1.52 ± 0.37 ^D,b^	1.93 ± 0.23 ^D,c^	2.33 ± 0.77 ^D,d^
CXU	Curry	42.23 ± 0.97 ^A,a^	49.9 ± 0.65 ^A,b^	51.63 ± 1.59 ^A,c^	54.47 ± 0.64 ^A,d^	55.76 ± 0.58 ^A,e^
	Coffee	3.95 ± 0.6 ^B,a^	9.31 ± 0.7 ^B,b^	9.8 ± 0.69 ^B,c^	12.01 ± 0.51 ^B,d^	11.91 ± 0.6 ^B,d^
	Wine	6.04 ± 0.6 ^C,a^	10.33 ± 0.36 ^C,b^	11.24 ± 0.82 ^C,c^	13.79 ± 0.7 ^C,d^	13.52 ± 0.51 ^C,d^
	Distilled water	1.38 ± 0.31 ^D,a^	1.66 ± 0.38 ^D,b^	2.13 ± 0.44 ^D,c^	1.82 ± 0.41 ^D,d^	2.18 ± 0.62 ^D,c^
CS	Curry	45.06 ± 2.76 ^A,a^	48.89 ± 0.77 ^A,b^	50.79 ± 0.76 ^A,c^	53.87 ± 0.62 ^A,d^	54.12 ± 0.7 ^A,d^
	Coffee	6.7 ± 1.38 ^B,a^	11.29 ± 0.72 ^B,b^	12.12 ± 0.8 ^B,c^	16 ± 0.78 ^B,d^	15 ± 0.54 ^B,e^
	Wine	13.37 ± 0.86 ^C,a^	15.87 ± 1.02 ^C,b^	17.35 ± 0.86 ^C,c^	19.48 ± 0.96 ^C,d^	19.92 ± 0.72 ^C,d^
	Distilled water	1 ± 0.22 ^D,a^	0.98 ± 0.24 ^D,a^	1.38 ± 0.23 ^D,b^	1.4 ± 0.27 ^D,b^	1.72 ± 0.47 ^D,c^
CD	Curry	49.78 ± 1.59 ^A,a^	52.88 ± 0.6 ^A,b^	55.64 ± 1.59 ^A,c^	57.63 ± 0.92 ^A,d^	58.23 ± 0.78 ^A,e^
	Coffee	6.78 ± 0.81 ^B,a^	13.4 ± 0.63 ^B,b^	13.87 ± 1.29 ^B,b^	15.96 ± 0.97 ^B,c^	16.97 ± 0.87 ^B,d^
	Wine	11.33 ± 0.88 ^C,a^	17.83 ± 0.49 ^C,b^	18.23 ± 0.89 ^C,b^	20.21 ± 0.66 ^C,c^	20.08 ± 0.68 ^C,c^
	Distilled water	0.9 ± 0.24 ^D,a^	0.89 ± 0.2 ^D,a^	1.22 ± 0.36 ^D,b^	1.29 ± 0.31 ^D,b,c^	1.35 ± 0.43 ^D,c^
DF	Curry	48.89 ± 0.86 ^A,a^	54.91 ± 0.38 ^A,b^	55.84 ± 1.3 ^A,b^	58.51 ± 0.54 ^A,c^	58.95 ± 0.61 ^A,c^
	Coffee	5.51 ± 0.65 ^B,a^	9.35 ± 1.1 ^B,b^	11.14 ± 0.78 ^B,c^	12.19 ± 1.24 ^B,d^	12.85 ± 1.39 ^B,d^
	Wine	9.46 ± 0.67 ^C,a^	14.97 ± 0.44 ^C,b^	16.18 ± 0.81 ^C,c^	19.18 ± 0.55 ^C,d^	19.06 ± 0.46 ^C,d^
	Distilled water	0.97 ± 0.2 ^D,a^	2.29 ± 0.9 ^D,b^	2.81 ± 0.68 ^D,c^	2.74 ± 0.59 ^D,b,c^	2.79 ± 0.68 ^D,b,c^
DXU	Curry	48.36 ± 1.12 ^A,a^	60.11 ± 0.7 ^A,b^	62.31 ± 1.36 ^A,c^	64.99 ± 0.66 ^A,d^	65.05 ± 0.68 ^A,d^
	Coffee	12.18 ± 1.59 ^B,a^	19.06 ± 1.49 ^B,b^	20.23 ± 1.09 ^B,b^	23.26 ± 1.33 ^B,c^	24.97 ± 1.08 ^B,d^
	Wine	16.45 ± 1.03 ^C,a^	28.18 ± 0.92 ^C,b^	31.47 ± 0.74 ^C,c^	35.62 ± 1.13 ^C,d^	36.32 ± 1.35 ^C,d^
	Distilled water	1.07 ± 0.33 ^D,a^	1.46 ± 0.4 ^D,b^	1.7 ± 0.36 ^D,c^	1.84 ± 0.34 ^D,c^	1.75 ± 0.49 ^D,c^
Z250	Curry	48.49 ± 0.77 ^A,a^	51.66 ± 2.5 ^A,b^	54.15 ± 2.41 ^A,c^	56.27 ± 2.51 ^A,d^	56.91 ± 2.25 ^A,d^
	Coffee	6.24 ± 1.2 ^B,a^	12.02 ± 0.42 ^B,b^	12.42 ± 0.45 ^B,c^	14.71 ± 0.5 ^B,d^	15.7 ± 0.7 ^B,e^
	Wine	8.71 ± 1.01 ^C,a^	15.14 ± 1.19 ^C,b^	17.09 ± 0.6 ^C,c^	20.46 ± 0.96 ^C,d^	21.1 ± 0.99 ^C,e^
	Distilled water	1.34 ± 0.5 ^D,a^	1.86 ± 0.75 ^D,a,b^	2.01 ± 0.35 ^D,b,c^	1.73 ± 0.46 ^D,a,c^	2.25 ± 0.46 ^D,b^
Z350	Curry	52.57 ± 1.25 ^A,a^	56.77 ± 2.44 ^A,b^	58.32 ± 2.43 ^A,b,c^	60.65 ± 3.38 ^A,c,^^d^	61.22 ± 3.19 ^A,d^
	Coffee	11.07 ± 0.64 ^B,a^	16.41 ± 1.29 ^B,b^	17.61 ± 0.96 ^B,c^	21.17 ± 0.96 ^B,d^	21.19 ± 1.04 ^B,d^
	Wine	17.39 ± 0.91 ^C,a^	26.55 ± 1.01 ^C,b^	29.98 ± 1.2 ^C,c^	35.85 ± 1.04 ^C,d^	37.76 ± 1.05 ^C,e^
	Distilled water	0.99 ± 0.17 ^D,a^	0.99 ± 0.37 ^D,a^	1.25 ± 0.28 ^D,b^	0.98 ± 0.27 ^D,a^	1.36 ± 0.37 ^D,b^
MP	Curry	44.47± 0.97 ^A,a^	50.28 ± 0.81 ^A,b^	53.24 ± 1.1 ^A,c^	56.36 ± 1.19 ^A,d^	56.59 ± 1.46 ^A,d^
	Coffee	4.36 ± 0.39 ^B,a^	8.24 ± 0.65 ^B,b^	8.72 ± 0.61 ^B,b^	11.1 ± 0.7 ^B,c^	12.5 ± 0.95 ^B,d^
	Wine	5.54 ± 0.49 ^C,a^	9.09 ± 0.41 ^C,b^	11.28 ± 0.73 ^C,c^	13.65 ± 0.41 ^C,d^	15.61 ± 0.38 ^C,e^
	Distilled water	1.21 ± 0.3 ^D,a^	1.3 ± 0.27 ^D,a^	1.75 ± 0.38 ^D,b^	2.45 ± 0.47 ^D,c^	2.32 ± 0.36 ^D,d^
TNC	Curry	46.77 ± 1.17 ^A,a^	52.38 ± 1.47 ^A,b^	56.18 ± 0.9 ^A,c^	57.4 ± 1.38 ^A,c,d^	57.88 ± 1.09 ^A,d^
	Coffee	7.81 ± 1 ^B,a^	12.57 ± 0.92 ^B,b^	13.44 ± 1.25 ^B,b^	16.78 ± 1.17 ^B,c^	16.23 ± 0.85 ^B,c^
	Wine	10.91 ± 0.73 ^C,a^	15.19 ± 0.53 ^C,b^	16.92 ± 0.6 ^C,c^	19.39 ± 0.81 ^C,d^	20.09 ± 0.78 ^C,e^
	Distilled water	0.71 ± 0.24 ^D,a^	0.89 ± 0.33 ^D,a,b^	0.93 ± 0.27 ^D,b^	1.04 ± 0.17 ^D,b^	1.56 ± 0.4 ^D,c^

Different uppercase letters represent statistically significant differences (*p* < 0.05) among colorants for a given material at the same timepoint. Different lowercase letters indicate statistically significant differences (*p* < 0.05) at different time points of a given material immersed in the same colorant.

**Table 3 materials-15-06710-t003:** Mean and SD of CIE L*, a*, and b* parameters of composite resins after immersion in solutions for 28 days.

Material	Solution	−∆L	∆a	∆b	∆E_5_
B2	Curry	−8.83 ± 1.21	5.93 ± 1.51	54.8 ± 1.16	55.85 ± 1.35
	Coffee	−19.2 ± 0.95	−1.56 ± 0.72	−6.28 ± 1.71	20.32 ± 1.22
	Wine	−20.94 ± 1	−4.8 ± 0.53	15.68 ± 1.167	26.63 ± 0.74
	Distilled water	0.55 ± 0.75	−0.73 ± 0.55	−1 ± 0.66	1.65 ± 0.66
F03	Curry	−6.71 ± 0.79	2.65 ± 0.78	47.66 ± 0.99	48.22 ± 0.97
	Coffee	−17.54 ± 0.84	−0.39 ± 0.46	−8.73 ± 1.23	19.55 ± 0.66
	Wine	−20.44 ± 0.70	−1.99 ± 0.86	16.40 ± 2.82	26.41 ± 1.52
	Distilled water	−0.19 ± 1.16	−0.52 ± 0.68	−1.33 ± 1.51	2.33 ± 0.77
CXU	Curry	−4.39 ± 0.77	−1.86 ± 0.57	55.55 ± 0.6	55.76 ± 0.58
	Coffee	−10.7 ± 0.67	3.81 ± 0.33	3.42 ± 0.98	11.91 ± 0.6
	Wine	−13.85 ± 0.85	2.2 ± 0.44	1.38 ± 0.87	13.52 ± 0.51
	Distilled water	0.53 ± 0.91	−0.29 ± 0.68	−1.63 ± 0.58	2.18 ± 0.62
CS	Curry	−7.14 ± 0.99	4.11 ± 0.91	53.47 ± 0.7	54.12 ± 0.7
	Coffee	−14.3 ± 0.59	3.14 ± 0.37	−2.82 ± 1.62	15 ± 0.54
	Wine	−19.8 ± 0.72	0.914 ± 1.13	−1.21 ± 1.14	19.92 ± 0.72
	Distilled water	0.33 ± 0.83	−0.88 ± 0.47	−0.97 ± 0.93	1.72 ± 0.47
CD	Curry	−4.97 ± 1.69	−0.61 ± 0.8	57.98 ± 0.82	58.23 ± 0.78
	Coffee	−16.84 ± 0.91	1.15 ± 0.39	−1.39 ± 1.06	16.97 ± 0.87
	Wine	−19.29 ± 0.59	−1.97 ± 1	4.98 ± 1.18	20.08 ± 0.68
	Distilled water	0.86 ± 0.58	−0.09 ± 0.3	0.19 ± 0.73	1.35 ± 0.43
DF	Curry	−4.84 ± 0.71	−0.88 ± 0.62	58.74 ± 0.61	58.95 ± 0.61
	Coffee	−12.22 ± 1.44	3.56 ± 0.23	0.11 ± 1.71	12.85 ± 1.39
	Wine	−18.19 ± 0.60	0.04 ± 0.61	5.24 ± 2.13	19.06 ± 0.46
	Distilled water	0.816 ± 0.91	−1.04 ± 0.35	2.26 ± 0.59	2.74 ± 0.59
DXU	Curry	−8.39 ± 0.63	5.94 ± 0.59	64.22 ± 0.67	65.05 ± 0.68
	Coffee	−24.3 ± 1.09	4.92 ± 0.23	2.82 ± 0.98	24.97 ± 1.08
	Wine	−30.5 ± 0.88	0.06 ± 0.87	19.64 ± 1.94	36.32 ± 1.35
	Distilled water	0.26 ± 0.8	0.02 ± 0.33	0.88 ± 1.24	1.75 ± 0.49
Z250	Curry	−5.73 ± 1.05	3.99 ± 1.39	55.41 ± 1.34	56.91 ± 2.25
	Coffee	−14.8 ± 0.8	5.01 ± 0.33	−1.08 ± 1.08	15.7 ± 0.7
	Wine	−21.22 ± 0.86	4.3 ± 0.44	2.1 ± 0.85	21.77 ± 0.89
	Distilled water	1.72 ± 0.67	0.29 ± 0.21	−1.15 ± 0.78	2.25 ± 0.46
Z350	Curry	−15.51 ± 1.24	17.43 ± 1.32	58.57 ± 0.89	61.22 ± 3.19
	Coffee	−20.90 ± 0.95	2.57 ± 0.7	1.72 ± 1.51	21.19 ± 1.04
	Wine	−30.03 ± 1.18	5.46 ± 0.72	22.17 1.34	37.76 ± 1.05
	Distilled water	0.57 ± 0.82	0.14 ± 0.31	−0.75 ± 0.69	1.36 ± 0.37
MP	Curry	−2.27± 0.61	−1.75 ± 0.55	56.93 ± 1.34	56.59 ± 1.46
	Coffee	−11.81 ± 0.85	3.8 ± 0.69	0.12 ± 1.48	12.5 ± 0.95
	Wine	−14.73 ± 0.39	3.34 ± 0.68	−3.82 ± 0.87	15.61 ± 0.38
	Distilled water	1.06 ± 0.76	−1.1 ± 0.21	1.46 ± 0.73	2.32 ± 0.36
TNC	Curry	−6.8 ± 0.98	2.39 ± 1.53	57.36 ± 1.11	57.88 ± 1.09
	Coffee	−15.41 ± 1.01	4.37 ± 0.49	−0.56 ± 2.54	16.23 ± 0.85
	Wine	−17.34 ± 0.87	3.95 ± 0.51	8.3 ± 1.67	19.7 ± 0.67
	Distilled water	0.3 ± 0.96	0.14 ± 0.33	−0.58 ± 1.1	1.56 ± 0.4

**Table 4 materials-15-06710-t004:** Means and SD of water sorption and solubility (μg/mm^3^) for the tested composite resins.

Material	Water Sorption	Water Solubility
B2	23.73 ± 0.54 ^d^	0.85 ± 0.49 ^a^
F03	30.11 ± 0.52 ^h^	0.89 ± 0.18 ^a^
CXU	16.58 ± 0.34 ^f^	0.31 ± 0.21 ^a^
CS	21.2 ± 0.42 ^e^	7.04 ± 1.79 ^c^
CD	20.94 ± 0.38 ^e^	5.7 ± 0.38 ^c^
DF	19.14 ± 0.49 ^g^	1.11 ± 0.49 ^a^
DXU	17.00 ± 0.17 ^b^	1.36 ± 0.57 ^a^
Z250	17.79 ± 0.41 ^b^	0.77 ± 0.17 ^a^
Z350	32.62 ± 1.21 ^c^	2.95 ± 0.51 ^b^
MP	14.32 ± 0.4 ^a^	0.51 ± 0.33 ^a^
TNC	18.00 ± 0.46 ^b,g^	1.03 ± 0.17 ^a^

Different lowercase letters indicate statistically significant differences (*p* < 0.05) among materials for a given property.

## Data Availability

The data presented in this study are available from the corresponding authors upon reasonable request.

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
