# Peer review of "Evaluation of the Color Stability, Water Sorption, and Solubility of Current Resin Composites"

_materials, 2022, doi:10.3390/ma15196710_

Round 1
Reviewer 1 Report
Dear author
The paper is very good many results but please see below my suggestions
Introduction is too long
For Water sorption and solubility equation please give a number and mention in text
Figure 1. need to be more visible at printing because all should be able to read after printing, Please include legend more close or inside of graph
Figure 3. legend is too small Please increase at the same size with the text of paper
Figure 4. on Y axist you write too small. A) and ) should be more close or inside of graphs
In comclusion please avoid numbers: 1. 2. 3. etc
Author Response
Dear reviewer:
We deeply appreciate your favorable consideration and the positive and insightful comments. Now we have revised the paper according to your comments. The revisions are addressed point by point below and modifications are marked with yellow background in the context.
Manuscript ID: materials-1914907
1) Introduction is too long.
Reply: We have simplified the introduction according to your comments.
2) For Water sorption and solubility equation please give a number and mention in text
Reply: We have given numbers to equations and mentioned them in the text.
3) Figure 1. need to be more visible at printing because all should be able to read after printing, Please include legend more close or inside of graph
Reply: We have modified Figure 1 according to your comments.
4) Figure 3. legend is too small Please increase at the same size with the text of paper
Reply: We have modified according to your comments.
5) Figure 4. on Y axist you write too small. A) and ) should be more close or inside of graphs
Reply: We have modified Figure 4 according to your comments.
6) In conclusion please avoid numbers: 1. 2. 3. Etc
Reply: We have deleted numbers: 1.2.3.
Reviewer 2 Report
In my opinion, the paper is an original scientific work that would be interesting for the readers of the journal.
Aim and hypothesis are clearly stated, materials and methods appropriate described. Results very precisely presented and discussed. Conclusions are well written.
I suggest to accept this manuscript for publication.
Author Response
Dear reviewer:
We sincerely appreciate for your positive comments and acceptance for this article.
Reviewer 3 Report
Dear authors,
the article covers a very interesting topic.
Nevertheless, I suggest some changes in order to improve the overall quality of the manuscript for the readers.
Line 45-6:
A very recent and complete review about staining and color stability should be cited.
Please add the following reference:
Paolone G, Formiga S, De Palma F, Abbruzzese L, Chirico L, Scolavino S, Goracci C, Cantatore G, Vichi A. Color stability of resin-based composites: Staining procedures with liquids-A narrative review. J Esthet Restor Dent. 2022 Apr 9. doi: 10.1111/jerd.12912. Epub ahead of print. PMID: 35396818.
Line 53:
Demarco et al. provided a systematic review of this topic. They stated that esthetics is one of the most relevant reasons for restoration replacement. For this reason, I suggest to cite it:
Demarco FF, Collares K, Coelho-de-Souza FH, Correa MB, Cenci MS, Moraes RR, Opdam NJ. Anterior composite restorations: A systematic review on long-term survival and reasons for failure. Dent Mater. 2015 Oct;31(10):1214-24. doi: 10.1016/j.dental.2015.07.005. Epub 2015 Aug 21. PMID: 26303655.
Line 104-107:
Please specify the temperature for every staining solution. If it is 37°C specify that was for every solution
Line 158-160:
The type of statistical analysis is not the best for this type of study.
When a specimen (the same specimen) is observed at several time intervals, an “ANOVA for Repeated Measures” test should be used. Please support your choice of this kind of test
The authors should mention that the study lacks of brushing simulation. Studies that include brushing simulators show less discoloration.
At the end of the discussion, please add the other limitations to the limitation paragraph: no brushing, no real temperatures, no repolishing effect (see below) etc.
The authors could have considered a repolishing procedure.
Specimens with a color change higher than the acceptability threshold could lower the DeltaE value below it.
Please mention this in the discussion.
Author Response
Dear reviewer:
We deeply appreciate your favorable consideration and the positive and insightful comments. Now we have revised the paper according to your comments. The revisions are addressed point by point below and modifications are marked with yellow background in the context.
Manuscript ID: materials-1914907
1) Line 45-6: A very recent and complete review about staining and color stability should be cited.Please add the following reference: Paolone G, Formiga S, De Palma F, Abbruzzese L, Chirico L, Scolavino S, Goracci C, Cantatore G, Vichi A. Color stability of resin-based composites: Staining procedures with liquids-A narrative review. J Esthet Restor Dent. 2022 Apr 9. doi: 10.1111/jerd.12912. Epub ahead of print. PMID: 35396818.
Reply: We have added this reference in the context according to your comments.
2) Line 53: Demarco et al. provided a systematic review of this topic. They stated that esthetics is one of the most relevant reasons for restoration replacement. For this reason, I suggest to cite it: Demarco FF, Collares K, Coelho-de-Souza FH, Correa MB, Cenci MS, Moraes RR, Opdam NJ. Anterior composite restorations: A systematic review on long-term survival and reasons for failure. Dent Mater. 2015 Oct;31(10):1214-24. doi: 10.1016/j.dental.2015.07.005. Epub 2015 Aug 21. PMID: 26303655.
Reply: We have cited this reference according to your comments.
3) Line 104-107: Please specify the temperature for every staining solution. If it is 37°C specify that was for every solution
Reply: In the new manuscript, We have described that 37°C was for every sample and every solution.
4) Line 158-160: The type of statistical analysis is not the best for this type of study. When a specimen (the same specimen) is observed at several time intervals, an “ANOVA for Repeated Measures” test should be used. Please support your choice of this kind of test
Reply: Thanks for your comments. We previously chose ANOVA for comparing statistical differences of color changes between materials at a given solution and a given time point; between time points at a given material and a given solution; and between solutions at a given material and a given time point. For this problem, we checked some literatures. Sincerely, “ANOVA for Repeated Measures” could be a better choice when a specimen (the same specimen) is observed at several time intervals. We found one reference, their statistical methods and study design were similar to ours. Thus, we performed “ANOVA for Repeated Measures” to evaluate the effects of immersion time according this article and rewrote statistical method in the new manuscript and renewed a few statistical results in Table2. Overall, these changes did not influence our results and conclusions.
Reference: Shin J W, Kim J E, Choi Y J, Shin S H, Nam N E, Shim J S, et al. Evaluation of the Color Stability of 3D-Printed Crown and Bridge Materials against Various Sources of Discoloration: An In Vitro Study. Materials (Basel), 2020, 13.
5) The authors should mention that the study lacks of brushing simulation. Studies that include brushing simulators show less discoloration. At the end of the discussion, please add the other limitations to the limitation paragraph: no brushing, no real temperatures, no repolishing effect (see below) etc.
The authors could have considered a repolishing procedure.Specimens with a color change higher than the acceptability threshold could lower the DeltaE value below it.
Please mention this in the discussion.
Reply: We have added these limitations in the end of the discussion according to your comments.